# Detection of Tennis Activities with Wearable Sensors

**DOI:** 10.3390/s19225004

**Published:** 2019-11-16

**Authors:** Luis Benages Pardo, David Buldain Perez, Carlos Orrite Uruñuela

**Affiliations:** Aragon Institute of Engineering Research, University of Zaragoza, Mariano Esquillor, 50018 Zaragoza, Spain; l_benages@hotmail.com (L.B.P.); buldain@unizar.es (D.B.P.)

**Keywords:** gyroscopes, accelerometers, tennis strokes, spectrograms, autoencoder, deep learning

## Abstract

This paper aims to design and implement a system capable of distinguishing between different activities carried out during a tennis match. The goal is to achieve the correct classification of a set of tennis strokes. The system must exhibit robustness to the variability of the height, age or sex of any subject that performs the actions. A new database is developed to meet this objective. The system is based on two sensor nodes using Bluetooth Low Energy (BLE) wireless technology to communicate with a PC that acts as a central device to collect the information received by the sensors. The data provided by these sensors are processed to calculate their spectrograms. Through the application of innovative deep learning techniques with semi-supervised training, it is possible to carry out the extraction of characteristics and the classification of activities. Preliminary results obtained with a data set of eight players, four women and four men have shown that our approach is able to address the problem of the diversity of human constitutions, weight and sex of different players, providing accuracy greater than 96.5% to recognize the tennis strokes of a new player never seen before by the system.

## 1. Introduction

The role of data in sport has grown significantly in recent years, being a subject of research in continuous development. Professionals and amateurs look for a way to obtain information about their performance or the possibility of improvement. Machine learning allows for the production of objective measures, it enables the detection of things which currently cannot be done by humans. Nowadays, the development of new sensors of smaller size and lower consumption allows data collection through portable or external sensors to be achieved much more easily, without interfering with the activities carried out. The information obtained by these sensors is of great interest in a wide variety of applications, such as medical applications for the monitoring of patients with diseases that need monitoring [1], in the rehabilitation of diseases or injuries by monitoring the progress of the patient [2,3,4], in the compilation of sports statistics for the visualization of the progress made [5], or in the improvement of technique in the performance of certain sports [6,7].

In recent decades, the state of the art in human activity recognition (HAR) has been expanding through the development of new techniques that allow greater accuracy in the classification of activities from sensors. This paper addresses several challenges that go beyond the simple temporal analysis of a sensor to determine the type of movement. So, the proposed solution applies deep learning techniques with convolutional neural networks (CNNs) of semi-supervised learning to extract information from the spectrograms of the signals collected by two portable sensor nodes. By extracting spectrogram features using CNN, the model becomes more robust to changes in weight, height, age or sex of the subjects under analysis. The use of CNN for feature extraction provides invariance in the position and deformation of the spectrogram. Therefore, although the same movement has different frequencies in the spectrogram when performed by different users, the forms obtained are the same, just shifted in frequency. So, the CNN may generate representations of features invariant to these changes.

Without loss of generality in sport analysis, this paper aims to recognise the activities carried out by a tennis player. The main goal is the development of a classification model to distinguish among a set of strokes and movements that can be performed during a tennis match. To collect significant information on the activities, two sensor nodes are used—one placed on the user’s wrist and the other placed on the waist, transmitting accelerometer and gyroscope information on its three coordinate axes.

Figure 1 shows the overall overview of the system designed to accomplish this goal. It is formed by three main modules—sensors, feature extraction and recognition. The tennis strokes under consideration are forehand, backhand, lob and volley.

The rest of the paper is divided as follows. Section 2 is dedicated to the state of the art. Section 3 introduces the way data have been collected. Section 4 describes the classifier. The design of the experimentation is shown in Section 5. Several experiments based on the dataset generated in this paper are conducted in Section 6. Finally, conclusions are discussed in Section 7.

## 2. State of the Art

Human Activity Recognition (HAR) is a field of research in continuous growth since the appearance of sensors capable of capturing information from the environment during the performance of any activity. In recent decades, a wide variety of methods for monitoring sports activities has appeared. One option is the detection of activities by treating video images [8] with cameras placed in the environment where the subject is located. However, this option is only suitable for certain types of applications since they are usually expensive and involve privacy problems and high computational costs in the treatment of images. The most common option when selecting sensors for HAR is the use of portable sensors, because the current technologies of these devices have low power consumption and small dimensions [9]. These benefits make them ideal for the collection of information for long periods and in any environment without interfering with the activities carried out.

In recent years, there has been great interest in the use of wearable technology in sport to improve performance and prevent injuries, see for example Reference [10] as a review. A significant advantage of these devices is the ability to monitor athletes in the field instead of inside a laboratory [11]. According to these reviews, the most important requirements in wireless sensor devices in sport are the quality of sensing, bit rate and communication range. Battery life and power consumption can be rectified by defining how the wearable is primarily going to be used. Wireless data transfer is a necessity, but signal loss must be minimized in order for data to be beneficial. Therefore, the choice of sensors and the communication protocol are crucial to the success of the application.

There are different types of wearable sensors used for HAR systems, such as accelerometers, gyroscopes, magnetometers, temperature sensors, barometers, locators, microphones, pulsometers or electrocardiographs. However, most applications researched in the state of the art use only accelerometers and gyroscopes, since the information they provide is sufficiently relevant for the classification of a large number of activities [12,13,14]. The place where the sensors are located in the body during the performance of the activities to be classified is also a design parameter in the research [15]. Changing the location of the sensor in the body causes the information obtained by the sensors to vary, so that the data obtained can be more or less representative of the activities performed. Using wearable sensors, the collection of the data obtained by the different sensors is usually carried out by means of wireless technologies, using as a master device a mobile smartphone or a laptop connected to different slave sensors through a low consumption protocol, such as for example Bluethooth Low Energy (BLE) or ZigBee [4,9].

Feature detection and extraction is one of the most important processes when developing a HAR system. Traditionally, hand-crafted feature descriptions, both in the temporal domain and in the frequency domain, have been used in HAR [1,6,15,16,17]. However, HAR models developed in this way present major problems of generalization [18,19]. Currently, researchers are trying to perform automatic extraction of relevant features from the input data. Thus, even if the set of input data changes, the model would still obtain characteristics sufficiently representative of the data to obtain good results in the classification. The most common technique for automatic feature extraction in HAR is based on the application of Deep Neural Networks (DNN), convolutional neural networks (CNN) being the most used [19].

CNNs are very popular in image processing as they automatically learn the necessary filters for pattern extraction [20,21]. In this way, the CNN learns discriminative features of the input data, presenting invariance before their position, scale and orientation. This quality makes them suitable for many applications of pattern recognition in images. This kind of networks are also used to extract features in temporal signals such as those provided by wearable sensors [22,23,24,25,26]. To do this, a 2D representation of said signals is usually carried out as if it were an image, so that the x and y coordinates of that representation are associated with pixel indices in an image [27]. There are different techniques to transform 1D signals to provide a 2D representation for the CNN. As an example, the temporal signal can be used directly as a 2D representation [24], stacking all the signals of the different sensors and applying the FFT [25] or perform the calculation of the spectrogram of the signals [23] that allows the extraction of the characteristics of frequency evolution over time. Another, less used, way to apply CNN to temporal signals is to obtain each dimension of a sensor as an independent channel (in the same way as done in an RGB image) and apply convolution in 1D to each channel [19]. In a recent work [28], the authors investigate golf swing data classification methods based on varieties of representative CNN which are fed with swing data from embedded multi-sensors, to group the multi-channel golf swing data labelled by hybrid categories from different golf players and swing shapes. In this article we use CNN applied in spectrograms, not 1D signals. In this sense, our approach is robust to player morphology.

Another kind of DNN used to extract representative features of the input data is the Autoencoder. This type of DNN is learning in its hidden layers an encoding of the input data by replicating the input at the output of the network, using unlabelled data. Once the network is trained, the coding obtained in the intermediate layer has enough information to generate an output of a particular class with which it has been trained. Thus, the intermediate layer of the Autoencoder contains a representative, and generally compressed, coding of the data introduced to the network. This coding is a representation similar to the feature vector obtained in CNNs, from which the output classifier is trained.

In relation to classification, HAR applications such as in References [12,13,29] use supervised (k-NN, MLP, or SVM ) as well as unsupervised (k-means or HMM) training. Traditionally, researchers opted more for supervised training methods because the activities to be classified were very limited and not very complex. However, to obtain a general model in a real HAR application, a large amount of labelled data is required for training the model, which is a slow and expensive process. Therefore, classifiers based on unsupervised or semi-supervised training methods are currently chosen. Semi-supervised learning allows the necessary amount of tagged data to be greatly reduced as long as a large amount of unlabelled data is available.

Nowadays, research in HAR is focusing on the use of DNNs for the classification of human activities. This is because, in addition to carrying out the classification tasks, it is a feature extractor of the input data. Some types of DNN allow semi-supervised training so that they learn high-level characteristics of a large amount of unlabelled training data, and subsequently learn to carry out classification work through a few labelled data.

For applications where no real-time processing is needed, the most common option is to perform off-line processing, with neural networks based on Deep Learning being the most commonly used classifiers today, as mentioned previously [18,21]. Deep Learning techniques can learn features of increasing complexity as they progress in their layers of neurons, so it is not necessary to manually design sets of specific features for the application. The classification is carried out in its last layers once high-level patterns have been extracted. The large number of connections between neurons that present this type of networks introduces the need for a very large number of parameters to be calculated. This requires the processing of a large number of training data, which entails a high computational cost.

All works above tackle the prediction task without taking into account information regarding transitions. Recently, some works exploits hierarchical group-based schemes to improve the classification efficiency and reduces the error through context awareness [30]. In this paper, we consider that the activities are pre-segmented, allowing for a future work to use a sliding window to carry out this process.

## 3. Data Collection

This section presents how to obtain 2D representation of 1D temporal signals. First, we discuss the collection of data by the sensor and then, the use of spectrograms to provide data to the recognition module.

### 3.1. Sensors

As mentioned before, the most commonly used sensors in HAR to date have been accelerometers and gyroscopes, since the information they provide is sufficient to recognize a large number of activities and they have a low cost.

One of the most relevant requirements for selecting the sensor is the power consumption, since sports analysis requires that the sensor battery has a range of several hours to collect data for long periods. Therefore, a device that has a low-power wireless technology is necessary, Bluetooth Low Energy (BLE) being the most used for the frequencies with which the accelerometers and gyroscopes work. Attending to these requirements this work uses the SensorTag CC2650STK from Texas Instruments. This sensor uses BLE low-power wireless technology so the battery can last from weeks to years, depending on the frequency of data transmission. In addition, it is capable of reprogramming using a debugger to obtain the desired sample rates.

In order to configure and collect the data from the two sensors at the same time from a computer, the Texas Instruments CC2540 USB Dongle has been used as an interface between the BLE protocol and a serial port. The control of this adapter is carried out using software provided by Texas Instruments called BTool. This software allows the USB device to act as a master in communication while the SensorTags function as slave devices. In systems with portable sensors, as used in this work, the sensors (slave devices) act as servers in two-way communication between two devices, since they are the ones that capture the information, while the USB dongle (master device) acts as a client in the data transmission by BLE, since it is the one that captures the requested data to the server. The BTool application allows the performance of different actions for communication, such as selecting the desired connection parameters, reading and writing features on the devices that act as servers, and so forth.

To collect relevant information about tennis activities, two sensor nodes (SensorTags) are used, one placed on the user’s wrist and the other placed on the waist, transmitting information of the accelerometer and gyroscope in its three coordinate axes. The total number of measured signals is 12 (3 axes of accelerometers plus 3 axes of gyroscopes for each SensorTag). The sampling frequency is 20 Hz, high enough to capture the movements made by a person without leading to a high consumption of the battery.

The SensorTag incorporates an MPU9250 motion sensor, which provides information on the three axes of accelerometer, gyroscope and magnetometer through the BLE communication protocol. To configure the motion sensor, the SensorTag firmware has the GATT (Generic Attribute Pofile) features in the BLE protocol with which the accelerometer and gyroscope sensors have been configured from a master device (the USB dongle with the BTool software). The accelerometer range has been set to ±16 G and the gyroscope range to ±2000 deg/s. The sampling frequency has been selected at 20 Hz.

To stream data packets captured by the sensors, the SensorTag has another GATT feature to enable or disable notifications in the BLE protocol. Enabling notifications in the BLE protocol allows slave devices to send sensed data packets directly to the master device without the master sending a data request. Each data packet is sent as soon as it is captured. In this way, it is possible to increase the speed at which the master device’s data is read. Therefore, the synchronization of the data received from the sensors has been performed off-line by means of the time stamp that each data has on the master device. In this way, the measurements coming from both devices were associated at the time the packets were received in the master device.

The BLE communication protocol allows a communication range of up to 100 m. However, for this application where high data transmission frequencies were needed, the maximum range used has been up to 20 m, since from this distance the data reception ratio began to decrease and did not meet the established sampling rates.

### 3.2. Spectrograms

In order to obtain a general activity classifier model for any user, the processing and segmentation of the sensor data has been carried out by calculating the signal spectrograms. By using spectrograms instead of the raw data of the sensors as input to a classifier, the differences in the characteristics of each user (age, weight, height, limb length, strength, etc.) have no influence on the results. This is because the form of the temporal evolution of the frequencies observed in the spectrograms will be the same during the performance of the activity, even though different users perform the same activity at different frequencies. Namely, the classifier will be able to distinguish the morphology of the spectrograms for a given activity regardless of whether it is displaced/stretched in frequencies or in time. An example of the spectrogram obtained by two different subjects running along the tennis court is shown in Figure 2. As can be noticed, both have similar forms, although the magnitudes of power recorded by one subject are greater than those of the other. Regardless of the intensity (related to the power) that one and the other represents, the system is able to recognize them as the convolutional neural network is insensitive to that difference in intensity.

The signal segmentation to compute the spectrograms is carried out by way of a sliding window along all the signals. It must be taken into account that the signal to be segmented must be of a single activity carried out by a user, otherwise an activity label could not be assigned to the spectrogram. In this way, the segmentation of the 12 signals from each axis of the sensors is performed in parallel. The length of each window is 20 samples, which corresponds to 1 second of data collection (sampling frequency of 20 Hz). The overlap between windows is 90% (18 overlapping samples). The Hann function is used to prevent aliasing when the discrete Fourier transform is calculated in each spectrogram window.

Taking into account the selected segmentation characteristics, the following parameters are chosen for the calculation of any spectrogram:
The number of signal samples covered by a spectrogram is 40, which is equivalent to 2 s of sensor data collection;The overlapping of consecutive sampling windows to calculate a spectrogram is 90% (36 superimposed samples);Following the Nyquist theorem the frequency resolution of the Fourier transform calculated for each window is set at half the sampling frequency of the sensors (10 Hz). Therefore, 11 values of the discrete Fourier transform are calculated, also taking into account the value of 0 Hz;The temporal resolution is equal to the number of windows obtained, which depends on the chosen parameters—number of samples per spectrogram, number of samples per window, and overlap between windows. In this case the number of windows is equal to 11;


With this parameter’s selection, spectrograms of dimension 11 × 11 are obtained. In Figure 2, two examples of the obtained spectrograms are shown, displaying them in two dimensions, where the third dimension, represented in a grey tone, corresponds to the magnitude calculated in the discrete Fourier transform. In this representation, each “pixel” of the spectrogram is the average of two contiguous data (so it has been represented as 10 × 10 images).

## 4. Classifier

To carry out the classification of tennis activities with the information provided by spectrograms, a semi-supervised learning approach based on two stages has been used, see Figure 3. In the first stage, an unsupervised learning model is constructed to learn representative features of the unlabelled data. It is convenient to have unlabelled data of the same type as those labelled so that the features learned are representative of the data to be classified. In the second stage, the trained model is used to obtain the features learned in the labelled data, so that a classifier model is trained with them in a supervised way.

The convolutional Autoencoder has been selected as an unsupervised learning model. Once the Autoencoder has been trained using the spectrograms of any sensor and activity, the first trained layers will be used to encode each of the 12 signals coming from the two sensors. Only one autoencoder has been trained with all the possible spectrograms (labelled or not) in the training set, so we can consider that it will behave as an universal extractor for these kinds of signals. Using the autoencoder as a feature extractor for any signal, we propose a modular architecture that can be expanded or reduced simply by replication/elimination of certain signal and its corresponding transformation by the autoencoder.

As shown in Figure 3, twelve parallel encoders process the spectrograms. The last convolutional layer of any encoder provides an output of 3 × 3 × 86 = 778, see Figure 4. Concatenating the twelve outputs of the encoders gives a vector of dimension 9288 that would be used as the input to a multilayer perceptron (MLP). The MLP will be trained in a supervised way to carry out the classifications of tagged activities.

### 4.1. Convolutional Autoencoder

Autoencoders are a kind of artificial neural network capable of learning efficient representations of input data, called encodings, without using labelled data. Therefore, these networks can be used as a powerful feature extractor for input data [31]. In this paper, a variant of this kind of networks has been used, which is called a convolutional Autoencoder. Its structure is the same as the Autoencoder but the fully-connected layers are changed for convolutional layers, see Figure 4.

By treating spectrograms as images, it is possible to determine features regardless of their scale and position. Thanks to this invariance, the trained model becomes robust to any user, since it does not matter if the shapes obtained in the spectrograms move/stretch in frequency or time.

Once the Convolutional Autoencoder has been trained, the initial encoding part of the autoencoder (conv 1, conv2 and conv3 in Figure 5) is used as the feature extractor for each signal, as depicted in Figure 4). After several trials, the chosen structure consists of three convolutional layers and one deconvolutional layer. The first layer includes 32 filters with kernels 6 × 6. The second layer includes 64 filters with 3 × 3 kernels and third layers were 86 filters with 3 × 3 kernels. The deconvolutional layer reconstructs the image with 32 filters with 6 × 6 kernels.

### 4.2. MLP

After training the Autoencoder, we obtain a feature extractor from the input data using its convolutional layers. This feature extractor is replicated twelve times (once per sensor axis), so that the characteristics obtained from each of them are introduced all together in a supervised classifier.

In this paper, three different classifier models have been developed to carry out different functionalities:
Tennis stroke classifier to know if the input data is a tennis stroke, as recorded in the dataset, or a normal activity;Normal activity classifier to identify seven no-tennis strokes, but carried out in a tennis match such as walking, running, jumping, bending down/lifting, standing, being seated and sitting/getting up;Classifier of tennis strokes to classify stokes in any of the ones collected in the database, that is, forehand, backhand, volley and lob.


## 5. Design of the Experimentation

As mentioned before, this work uses a couple of SensorTag CC2650STK from Texas Instruments, one in the wrist and the other one in the waist. The communication to the computer is made by the Texas Instruments CC2540 USB Dongle, as depicted in Figure 5.

Next, the dataset generated for the experimentation is introduced. Additionally, the selection of some parameters used in the training of the DNNs are commented.

### 5.1. Database

As the final goal is to identify the type of strokes and movements accomplished in a tennis match, a set of these actions has been recorded from four males and four females with the following characteristics, shown in Table 1.

As tennis strokes, the subjects performed forehand, backhand, volley and lob strokes as well as some usual activities such as walking, running, jumping, bending down/lifting, standing, being seated and sitting/getting up. Table 2 shows the number of samples per sensor axis in the whole database.

The table contains information on both, the number of raw data samples collected by each axis of the sensors and the number of spectrograms obtained from these data. The number of data per axis has been represented, since each of the 12 axes (3 accelerometers and 3 gyroscopes of each SensorTag) corresponds to the measurement at the same time, so they will then be introduced together to the classifier model.

The main problem with deep learning is the large amount of tagged data needed for training. To overcome this problem, we use the convolutional autoencoder, in which many of the time sequences recorded during a tennis match are provided to the system without any labelling process. In this sense, the convolutional autoencoder obtains a generic model of all types of activities that are carried out in a match. The first row in Table 2 shows the amount of unlabelled activities used to train the convolutional autoencoder. Afterwards, some new sequences are recorded and labelled by hand. They are used to train the MLPs that finally identify the kind of action and tennis stroke.

On the other hand, as the main objective of this work is to obtain a robust model regardless of the variability of the player who performs the activities, two different types of tests have been carried out, see Table 3.

TEST1: we consider all players except 1 in Table 1. For any of these players, 10 s are selected in the middle of the time sequence of each activity for the test. These 10 seconds correspond to 200 samples of raw data, that is, 41 spectrograms. The rest of the data is used for training. In this TEST1, the same players are used to train and test.

TEST2: all data of player 1 to test. The selected user is the first in Table 1, as it is the average morphological player among all users. This set is used to verify the effectiveness of the model developed in a real situation, that is, to assess its ability to deal with a new player never seen before. For training, we use the same data as in TEST1.

### 5.2. Convolutional Autoencoder Training

First of all, it is worth mentioning that the autoencoder has been trained with all the possible spectrograms (labelled or not) from the training set, Table 2.

Following Reference [31], the next design parameters are chosen:
As an activation function of the convolutional layers, the Exponential Linear Unit (ELU) has been chosen because it tends to converge cost to zero faster and produce more accurate results.He initialization is followed to initialize the weights of the convolutional layers.The cost function used is the Mean Squared error (MSE), calculated by means of the difference between the input to replicate and the reconstruction obtained at the network exit.The training algorithm used to adjust the weights of the network is the descent by the gradient together with the Adam optimization technique, since this technique accelerates the training of the network by being much faster than the descent by the gradient.


Training parameters:A training factor of 0.0001 has been selected with which the algorithm converges in a reasonable number of epochs.Mini-batch training has been carried out to improve the results and accelerate the training. The selected batch size has been 96 spectrograms (8 of each of the 12 sensor axes).The data at the input layer of the network is normalized with the mean and standard deviation of all training data.The number of cycles executed during training is 100.

### 5.3. MLP-Based Classifiers

As mentioned before, there are three MLP-based classifiers, one to classify tennis strokes from normal activities; another one to classify among the normal activities one of the eight options under consideration; finally a tennis stroke classifier to identify among the four strokes registered.

#### 5.3.1. Tennis versus No Tennis MLP

Figure 6 shows the architecture of the binary classifier stroke versus no-stroke. The input layer of the classifier has 9288 neurons and the output layer is formed by a single binary neuron, so that the output value will be close to one when the input data belongs to one of the tennis strokes and a zero in the case where it is any other type of activity. To select which data has a value close enough to 1 to indicate it as a tennis stroke, a threshold value is established as a design parameter with which the precision and recall results obtained can be modified.

Training parameters:
He initialization is followed to initialize all weights.The activation function of the input layer is the ELU function.Regarding the output neuron, the sigmoid activation function is used, so that the output of the neuron will exhibit a value between 0 and 1.The cost function chosen is cross entropy. The gradient descent technique has been used together with the Adam optimizer with a training factor of 0.0001.Input data normalization is performed by calculating the mean and standard deviation of the entire training set.Mini-batch training is applied. The number of training cycles is 35.


#### 5.3.2. Normal Activities MLP

An MLP has been developed to classify the data into one of the following 7 daily activities—walking, running, jumping, bending down, standing up, being seated and sitting. These activities were selected because they can happen on a tennis court, either between tennis strokes or on a break. Figure 7 shows the architecture for this classifier.

Training parameters:
To initialize all weights the He initialization is followed.The activation function of the input layer is the ELU function.The output layer consists of neurons to which the Softmax activation function is applied, therefore the probability that the input data belongs to that class is obtained in each output neuron. Softmax is chosen because a previous MLP has been trained to classify between tennis or no-tennis strokes, therefore, in this new MLP an exhaustive classification is carried out.The cost function chosen is cross entropy. The gradient descent technique has been used together with the Adam optimizer with a training factor of 0.0001.Input data normalization is performed by calculating the mean and standard deviation of the entire training set.Mini-batch training is applied. The number of training cycles is 120Neurons in every layer: 9288-35-7.


The MLP has as many neurons in the input layer as concatenated outputs of the same convolutional autoencoder applied in the 12 signals, that is, 9288 neurons. The output layer has 7 neurons, since it is the number of activities to classify, the predicted class being the one whose neuron has the greatest value at its output. The estimation of the number of hidden neurons is an important factor that determines the final accuracy of the classifier. In the case of choosing fewer neurons than necessary to separate the distributions of data of different classes, the erroneous classifications will increase. In the case of selecting too many hidden neurons, we increase considerably the number of parameters to be calculated in the network, and the risk of over-fitting the network to training data is increased.

Several architectures with a different number of hidden neurons were trained. Afterwards, 10 cross validations were carried out. As can be noticed in Figure 8, the network with the best results corresponds to 35 hidden neurons, since it is the one with the least dispersion of the obtained precision (exhibiting a more reliable behaviour) with acceptable precision. The architectures with a greater number of hidden neurons do not show a significant increase in the precision but they strongly increase the number of parameters of the network, so they are not of interest due to their greater tendency to over-fitting.

#### 5.3.3. Tennis Strokes MLP

An MLP was designed to classify four tennis strokes (right, backhand, volley and lob) plus one rejection class (any of the above). This class was taken into account to allow the system to be updated with more strokes in the future, such as service, and so forth.

The output of any neuron provides a value between 0 and 1 (sigmoid activation function is used), proportional to the individual probability of belonging to that class. In this way, the output tag corresponds to the neuron with the highest value at its output, as long as this value exceeds a certain threshold. If a class is selected as the most likely, but does not exceed the threshold value, it is assigned to the rejection class. A threshold value equal to 0.05 (5%) has been selected, which means that we rely heavily on the classification model.

Training parameters:He initialization is followed to initialize all weights.The activation function of the input layer is the ELU function.For the output neuron, the sigmoid activation function is used, so that the output of the neuron will exhibit a value between 0 and 1.The cost function chosen is cross entropy. The gradient descent technique has been used together with the Adam optimizer with a training factor of 0.0001.Input data normalization is performed by calculating the mean and standard deviation of the entire training set.Mini-batch training is applied. The number of training cycles is 45.Neurons in every layer: 9288-18-5.


The MLP has as many neurons in the input layer as concatenated outputs of the same convolutional autoencoder applied in the 12 signals, that is, 9288 neurons. The output layer has 5 neurons, 4 for tennis stroke classes, and one for the rejection class. As done before, several architectures with different numbers of hidden neurons have been trained. Afterwards, 10 cross validations have been carried out. As can be noticed in Figure 9, the network that presents the best results is that of 18 hidden neurons.

## 6. Results

This section shows some results regarding the three types of classifiers used in this paper, that is, binary classifiers between tennis and non-tennis activities; recognition of normal activities (between seven classes) and recognition of tennis stroke (between five classes).

Two types of tests were carried out; on the one hand, TEST1 where the data test set was separated from the training set and on the other hand, TEST2 where the test set is formed with the player not introduced in training.

Firstly, spectrograms are used individually, that is, like a single image. Since the classifier is dealing with temporal sequences, a median temporal filter is used. The final label of the pre-segmented sequence corresponds to the most voted label of all individual spectrograms.

In relation to the binary classifier, some metrics, such as ROC, Accuracy versus Recall and confusion matrix, are used to evaluate the performance of the classifier. For multi-class classifiers, the confusion matrix and the Kappa index are used. The Kappa index provides a value between 0 (no match) and 1 (perfect match), being a value greater than 0.7 representative of a good classifier.

### 6.1. Tennis versus No-Tennis

The architecture of this classifier is shown in Figure 6. The binary output relies on the appropriate selection of a threshold. To select it, the Precision-Recall (PR) curve and the ROC curve have been obtained, taking into account the training data set, see Figure 10.

The ROC curve shows the correct behaviour of the trained classifier, since the area under the curve is practically the unit. By observing the PR curve, it can be noticed that a similar score in accuracy and recovery is achieved with a threshold value of 0.5. However, it is preferable that all data related to tennis strokes are classified as such, so that more importance is given to obtaining a greater value of Recall. In this way, a threshold value of 0.4 is chosen to obtain a recall of 99.2%.

Figure 11 shows the results obtained in the classification by the tennis strokes filter model with the two test sets mentioned in the design of experimentation section.

Due to the chosen threshold, the accuracy value obtained for the test data is very high; most tennis strokes are classified as such. Although several data that are not tennis strokes are classified too, the tennis stroke MLP may also discard them.

Once the data is classified, only those that have been predicted as “Data_Tennis” will be entered into the tennis stroke MLP. In this way, 9 false positive and 654 true positive will be input in TEST1, the data test set separated from the training set, and 29 false positive and 6022 true positive in the case of TEST2 regarding the player not introduced in training.

In Figure 11, TEST2 shows the good generalization made by the model since the classification of the data of a subject not seen during the training shows better results even than the test set separated from the training data.

### 6.2. Normal Activity Classifier

In Figure 12, TEST1 shows the confusion matrix obtained with the test set considering data separated from the training set. The value of the Kappa index obtained is 0.9543, whilst Precision is 0.9609 and Recall is 0.9609.

The confusion matrix shows how most of the highlighted cells are on the main diagonal, pointing out the good performance of the classifier. In each row of the matrix, both the number of data in each predicted class and the percentage corresponding to the total data of the true class have been represented. In addition, a Kappa index close to one also shows the goodness of the classifier.

In Figure 12, TEST2 shows the classifications obtained from the activities carried out by a subject that was not incorporated into the training database. This test allows us to check the generalization capacity of the model to any new player. As expected, the most confusing classes are the class “bending down” with the class “sitting” due to the similarity of the activities and the class “standing up” with the class “being seated” because, being static activities, the frequency treatment of signals with spectrograms will not contribute differences between them.

### 6.3. Tennis Strokes Classifier

Figure 13 shows some results for the recognition of tennis strokes. It is worth mentioning that only these data that have passed the tennis versus non-tennis filter is applied to the MLP classifier. For TEST1, it is observed that the classification is practically correct in its entirety, except for a small percentage of lobs that have not been recognized as a tennis stroke.

Figure 13 shows TEST2 corresponding to the data of the subject not introduced to the MLP. It can be noticed that the data not belonging to any of the tennis strokes (which had been incorrectly classified by the previous filter) are now correctly classified. The rest of the strokes are classified with accuracy levels above 97%, except for the lob class, which has an accuracy of 91.6%. A lob in tennis is hitting the ball high and deep into the opponent’s court. The confusion of this class may occur because the user did not perform the lobs correctly, thus being confused with other movements. In any case, the MLP provides a Kappa index of 95.39%, which demonstrates the robustness of the model to deal with new players.

## 7. Conclusions

With professional sports teams and amateur people who place greater emphasis on technology and data, a vast field of research for data analysis has recently emerged to provide information to athletes for success and victory. Today, there is a great demand in the market for portable devices capable of gathering important information during the performance of any activity. This information is processed to obtain statistics or feedback to improve the sporting technique of the users.

This paper presents a classification system of tennis strokes that can address the diversity of body dimensions, weight and sex of different players. The system is based on accelerometers and gyroscopes attached to the wrist and waist of the subject. According to our knowledge, there is not any dataset available to work with, so the first step has been to develop a dataset using eight different subjects (gender, age, weight and height). The processing of the temporal data sequences of the sensors was carried out through the calculation of spectrograms that allow elimination of the dependencies related to the physical characteristics of the subjects.

The system was developed following a semi-supervised approach using TensorFlow libraries. Firstly, a Convolutional Autoencoder was selected to perform feature extraction in an unsupervised way. This unique autoencoder behaves as an universal feature extractor for each signal, building a modular architecture that can be expanded or reduced simply by replication/elimination of certain signal and its corresponding transformation by the autoencoder. Afterwards, an MLP was used for each classification step. Three different classifiers were developed in this work. The first one identified tennis versus non-tennis strokes. That data passing through this filter is further classified into one of the four tennis strokes, in case of tennis activities, otherwise in any of the seven normal activities or the rejection class. An average accuracy of 99.25% has been obtained for tennis stroke classification using a test database with people who had already been seen by the model and 96.51% for a new subject. These results clearly show the good performance and robustness of the classification system.

The current dataset has been obtained thanks to eight amateur players. For future work, we intend to extend the dataset to other kinds of tennis strokes using professional players, which can execute the actions in a more complex and extensive way. In addition, it would be interesting to collect data on-line to be able to compile statistics of strokes in real time.

Another possible long-term future line would be the synchronization of the data obtained by the sensors with a video capture system located in the environment capable of tracking the player’s skeleton. In this way, not only stroke statistics could be obtained, but also information on the execution technique of the movement performed. Therefore, the player could get feedback on each stroke to improve his/her level of play.

## Figures and Tables

**Figure 1 sensors-19-05004-f001:**
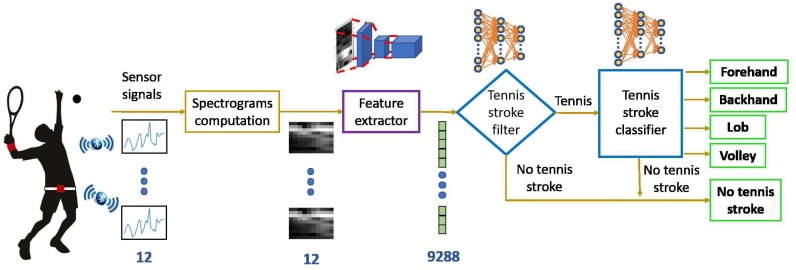
Detection of tennis strokes with wearable sensors and deep learning.

**Figure 2 sensors-19-05004-f002:**
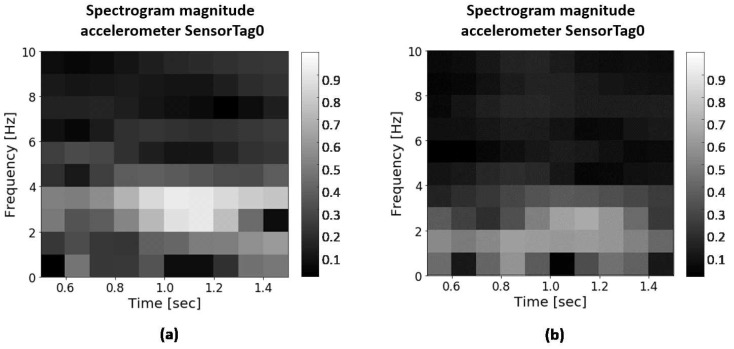
Spectrograms for running activity. (**a**) Male player. (**b**) Female player.

**Figure 3 sensors-19-05004-f003:**
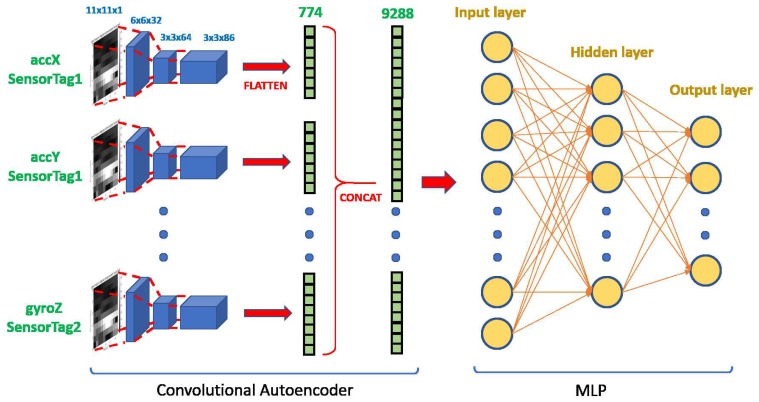
The complete classifier architecture.

**Figure 4 sensors-19-05004-f004:**
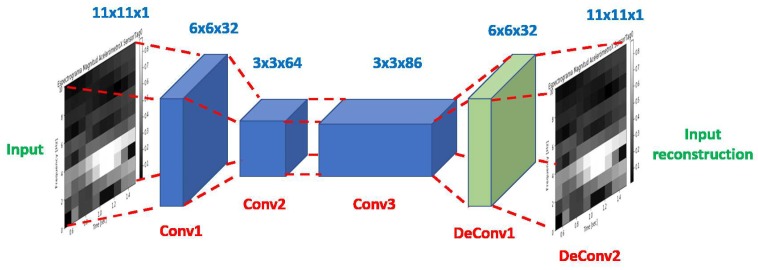
Convolutional Autoencoder Architecture used in this paper.

**Figure 5 sensors-19-05004-f005:**
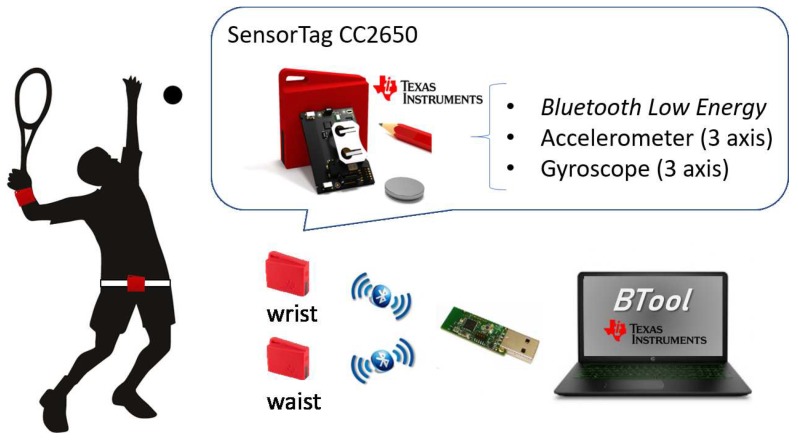
Sensors configuration.

**Figure 6 sensors-19-05004-f006:**
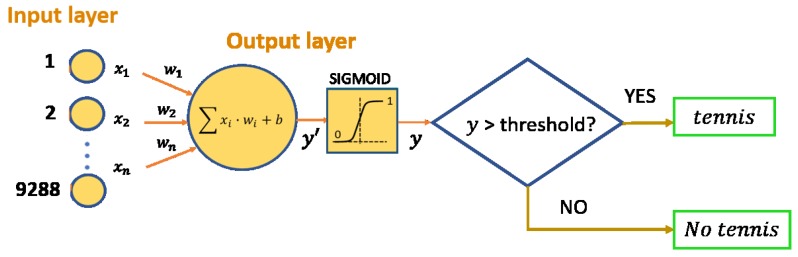
Tennis vs. no-tennis stroke MLP architecture.

**Figure 7 sensors-19-05004-f007:**
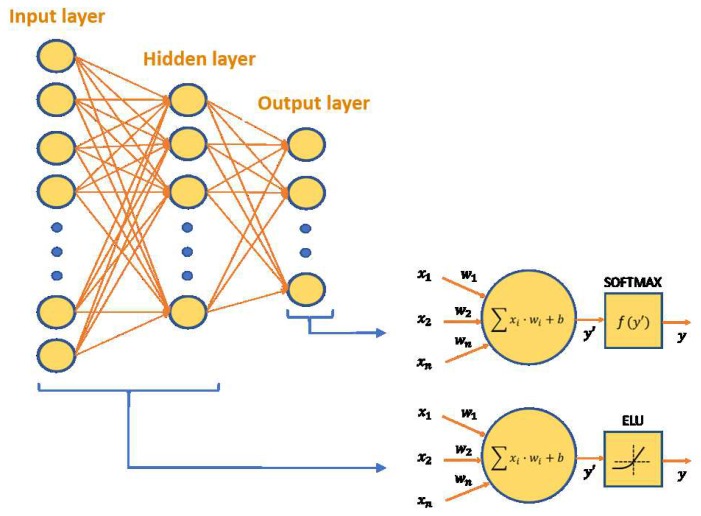
MLP architecture.

**Figure 8 sensors-19-05004-f008:**
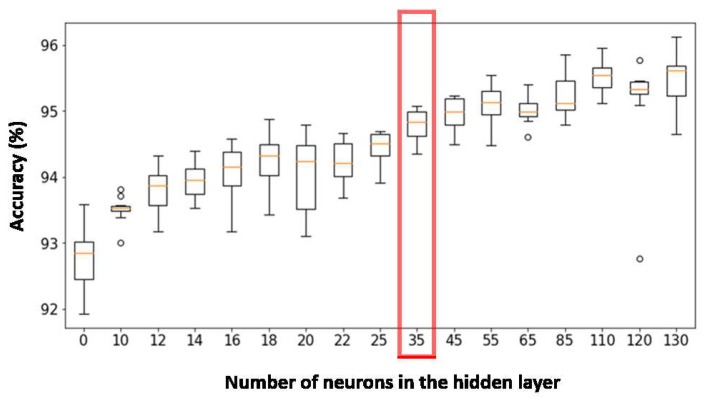
Accuracy obtained for every activity classifier architecture with 10 cross validations.

**Figure 9 sensors-19-05004-f009:**
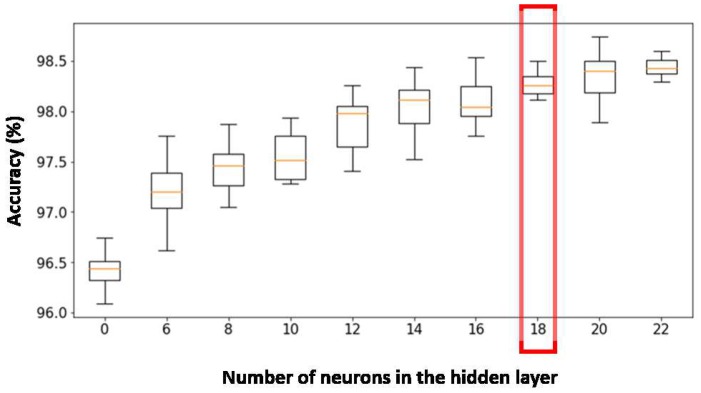
Accuracy obtained for every stroke classifier architecture with 10 cross validations.

**Figure 10 sensors-19-05004-f010:**
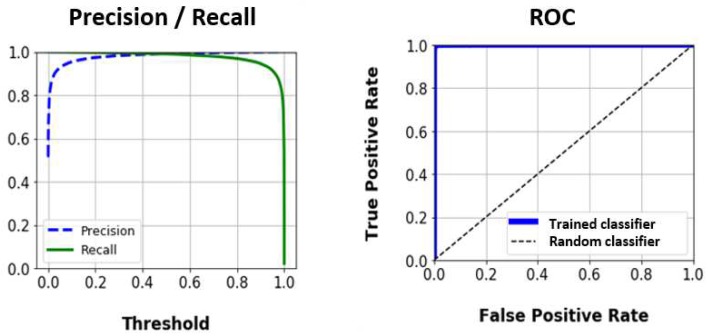
Precision-Recall curve and ROC curve of tennis strokes filter.

**Figure 11 sensors-19-05004-f011:**
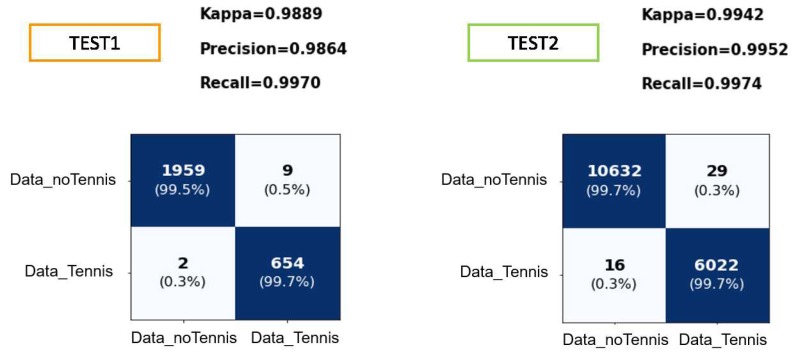
Confusion matrix for the tennis vs non-tennis activities. TEST1: Data test set separated from training set. TEST2: Test set regarding the player not introduced in training. Rows true label and columns the predicted label.

**Figure 12 sensors-19-05004-f012:**
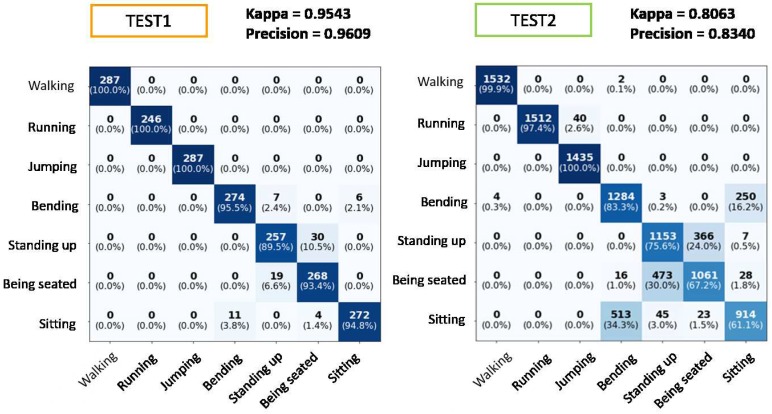
Confusion matrix for normal activities classification. TEST1: Data test set separated from training set. TEST2: Test set regarding the player not introduced in training. Rows true label and columns the predicted label.

**Figure 13 sensors-19-05004-f013:**
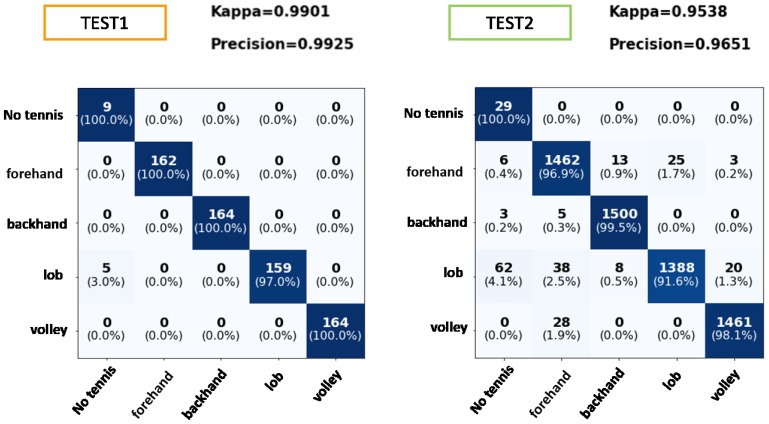
Confusion matrix for tennis strokes classification. TEST1: Data test set separated from training set. TEST2: Test set regarding with the player not introduced in training. Rows true label and columns the predicted label.

**Table 1 sensors-19-05004-t001:** Characteristics of the subjects in the database.

Subject	Gender	Age	Height (cm)	Weight (kg)	Laterality
1	M	23	174	73	right
2	M	24	190	80	right
3	M	21	182	76	right
4	M	30	168	70	right
5	F	23	160	65	left
6	F	27	164	59	right
7	F	23	159	57	right
8	F	19	169	62	right

**Table 2 sensors-19-05004-t002:** Number of samples for each sensor axis in the database.

Activity	No. of Samples	No. of Spectrograms
	Per Axis	Per Axis
No labelled	577,097	144,201
walking	49,791	12,373
running	42,679	10,605
jumping	48,387	12,022
bending down	48,702	12,100
standing up	49,845	12,386
being seated	50,847	12,637
sitting	48,654	12,089
forehand	30,085	7475
backhand	30,203	7504
volley	30,261	7518
lob	30,135	7488
TOTAL	1,036,686	258,398

**Table 3 sensors-19-05004-t003:** Number of spectrograms for training, TEST1 and TEST2.

Activity	Trainning	TEST1	TEST2
walking	10,552	287	1534
running	8807	246	1552
jumping	10,300	287	1435
bending down	10,272	287	1541
standing up	10,573	287	1526
being seated	10,772	287	1578
sitting	10,307	287	1495
forehand	5802	164	1509
backhand	5832	164	1508
volley	5834	164	1520
lob	5823	164	1501
TOTAL	94,874	2624	1669

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
