# Peer review of "Detection of Tennis Activities with Wearable Sensors"

_sensors, 2019, doi:10.3390/s19225004_

Round 1

Reviewer 1 Report

Accepted

A deep learning scheme based on feature extraction and deep learning is proposed to deal with the tennis related HAR. The raw sensor data are transformed into spectrograms and then the convolutional-pooling-autoencoder block is explicitly established as feature extractor before the MLP model. The whole work is interesting and some useful results have been given out in this paper. The following are the reviewer’s comments for the final publication.

To comprehensively characterize the feature engineering and HAR schemes, the following works should be discussed and compared with this manuscript.

[1] CCFS: An Confidence-based Cost-effective feature selection scheme for healthcare data classification, IEEE/ACM Transactions on Computational Biology and Bioinformatics, 2019. DOI 10.1109/TCBB.2019.2903804.

[2] GCHAR: An efficient Group-based Context–aware Human Activity Recognition on smartphone, Journal of Parallel and Distributed Computing, Volume 118, Part 1, August 2018, Pages 67-80.

The Figure 2. (in 5 of 17) should be re-drawn under the same standard. That is, if the colors standing for each ‘’pixel’’ are the same in two pictures, it is going to be easier to find the similarity.

Reviewer 2 Report

The manuscript describes the system that can be used for the distinguishing of four different tennis strokes from other activities in a tennis match, such as for example walking, running and sitting. The methods used for that are innovative and results are good. Activity classification results are very good for the tennis strokes and moderate for other activities. The structure of the paper and its presentation are adequate, but the paper needs a number of improvements before being suitable for the publication.

Major comments:

The title of the paper and its contents are not in accordance. The research is done on the sport-specific movements in tennis, therefore the word tennis must be included into the title. Even though the authors do the classification of general human activities, such as walking and running, they are done under the specific conditions during the tennis and they cannot be generalized without further testing. The assumption and/or requirement for low power consumption of sensor devices is inaccurate, if not wrong.

While that is true for a traditional sensor network, in sport we have easy access to all sensor devices and we can regularly change and/or charge their batteries. Therefore the more important requirements in wireless sensor devices in sport is the quality of sensing, bit rate, and communication range.  Authors should revise the text in Section 3.1 and perhaps include some references that deal with challenges of wireless communication of wearable sensor devices in sport.

While 20 Hz sampling frequencies is enough for the human gait acquisition, it is not adequate for moderate and high speed movement in sport, such as in tennis. I suggest reading and adding references that deal with the validation of inertial sensors (gyroscopes and accelerometer) for applications in sport. I am not arguing that 20 Hz is not enough for your application, but higher sensor sampling frequencies would probably give more accurate results when processed appropriately.

State of the art is not adequately covered. You should add the discussion of other sport-specific uses of CNN, such as for example in golf for different swing types classification.

CNN can work on 1D data, explain why you transform the 1D data to 2D data and then process it?

The explanation of the database and data is insufficient. Please add deeper explanation about the sample labelling and inclusion of the samples in Test 1 and Test 2. Also explain how the labelling process worked, etc.

Minor comments:

Add the explanation of the signal transmission mode (packet, streaming) from the sensors.

Discuss the synchronization method of the signals from the two independent sensor devices.

Give the communication range of the used sensor devices.

There are a lot of spelling mistakes where He should be replaced by The.

In figures 11 to 13 mark which dimension is the label and which dimension is the predicted result.

Give a comment on the possibility of activity recognition in real-time, during the match and how could it be used in tennis training.

Reviewer 3 Report

The paper is devoted to the analysis of a very interesting topic related to the use of different sensors for evaluation of motion during sport activities. I have the following specific comments to the present submission:

Major comments:

Abstract: I suggest to include here the specification of the proposed methodology and selected numerical results with the accuracy achieved during the classification process.

Page 5, Section 3.2: The methodology should be described into more details. Spectrograms in Fig. 2 have 10 by 10 elements and in the text the description of 11 by 11 elements is presented.

Page 6, Fig. 3: the convolution autoencoder should be better presented and separate steps of processing of the spectrograms should be explained into more details. It should be explained how the number 9288 was calculated (using 12 axis?)? The two layer network was used only? Which transfer functions were applied?

Page 6, row 2 from the bottom: The invariance of spectrograms to the scale is obvious. It should be explained why invariance to the position exists as well.

Page 7, Section 5: position of sensors is very important. How their locations was selected? Would it be possible to obtain better results with more sensors? Technical details of data acquisition and transmission should be presented as well. Was the sampling rate constant? In which format the data were acquired? Was the time synchronization of sensors sufficient?

Page 9, Section 5.3: It should be described which software tools were used for data classification? Were results for classification by different methods compared?

Page 10, Fig. 7: It is necessary to appreciate the use of softmax function. Its behaviour should be briefly mentioned in the text (with links to references)

Pages 16, References: I suggest to add a few further links related to methods proposed, further sensors use and possible physiological data acquisition and processing (which might be mentioned in the introduction as well) including for instance

[1] Kristo M, Ivasic-Kos M.: An overview of thermal face recognition methods, 41st International Convention on Information and Communication Technology, Electronics and Microelectronics (MIPRO),  2018, pp. 1-6  

[2] Procházka A., Charvátová H., Vyšata O., Kopal J., Chambers J.: Breathing Analysis Using Thermal and Depth Imaging Camera Video Records, MDPI: Sensors,  2017, 17, 1408, pp. 1-10  

Minor comments:

Several formal improvements should be done: Titles of Subsection 6.1,… should start with capital letters

All abbreviations should be in the list on page 16 (including ELU)
